# A Review of Natural Gas Hydrate Formation with Amino Acids

Bo Li [1,2,*], You-Yun Lu [1,2] and Yuan-Le Li [1,2]

1   State Key Laboratory of Coal Mine Disaster Dynamics and Control, Chongqing University, Chongqing 400044, China
2   School of Resources and Safety Engineering, Chongqing University, Chongqing 400044, China
*   Correspondence: libo86@cqu.edu.cn

**Abstract:** Natural gas is a kind of low-carbon energy source with abundant reserves globally and high calorific value. It is cleaner and more efficient than oil and coal. Enlarging the utilization of natural gas is also one of the important ways to reduce carbon emissions in the world. Solidified natural gas technology (SNG) stores natural gas in solid hydrates, which is a prospective, efficient, safe and environmental-friendly strategy of natural gas storage and transport. However, the slow growth rate and randomness of nucleation during natural gas hydrate formation in pure water hinder the industrial application of this technology. As a kind of new and potential additives, biodegradable amino acids can be adopted as favorable kinetic promoters for natural gas hydrate synthesis. Compared with other frequently used chemical additives, amino acids are usually more friendly to the environment, and are capable of avoiding foam formation during complete decomposition of gas hydrates. In this paper, we have reviewed the research progress of gas hydrate generation under the promotion of amino acids. The formation systems in which amino acids can enhance the growth speed of gas hydrates are summarized, and the impact of the concentration in different systems and the side chains of amino acids on hydrate growth have been illustrated. The thermodynamic and kinetic behaviors as well as the morphology properties of hydrate formation with amino acids are summarized, and the promotion mechanism is also analyzed for better selection of this kind of potential additives in the future.

**Keywords:** natural gas hydrate; amino acids; additives; kinetics; morphology

## 1. Introduction

Due to the rapid economic and social development of mankind and the continuous global industrialization, the energy demand of our world is anticipated to rise prominently in the next decades. With the desire of green and low-carbon society, the problem of pollution caused by huge amounts of contaminant emissions has been a major focus of all human beings during the consumption of coal and oil. Since the major constitution of natural gas is methane, which has a relatively high hydrogen to carbon ratio, its carbon dioxide emissions during combustion are also much lower [1]. Thus, it is generally recognized as a clean and low-carbon energy source.

Both in conventional and unconventional hydrocarbon sources such as natural gas hydrate, tight gas, shale gas, and coalbed methane [2–5], natural gas resources are widely distributed all over the world, while the distribution of reserves is relatively concentrated. In 2020, the global recoverable natural gas reserves are estimated to be $188.1 \times 10^{12}$ m$^3$, and about 70.4% are distributed in the Middle East and CIS countries (Figure 1a) [6]. The top five countries in the world with recoverable reserves of natural gas are Russia ($37 \times 10^{12}$ m$^3$), Iran ($32 \times 10^{12}$ m$^3$), Qatar ($25 \times 10^{12}$ m$^3$), Turkmenistan ($13.6 \times 10^{12}$ m$^3$) and the United States ($12.6 \times 10^{12}$ m$^3$), accounting for 64% of the world's total recoverable reserves [6]. The amount of recoverable natural gas of China is about $8.4 \times 10^{12}$ m$^3$, which ranks sixth globally (Figure 1b) [6]. Based on the current natural gas reserves-to-production ratio in 2020, natural gas can still be produced for about 48.8 years [6]. In addition, the amount of

carbon constrained by gas hydrates is approximately twice as large as that of traditional fossil fuels [7], which can provide additional supply as natural gas sources in the future.

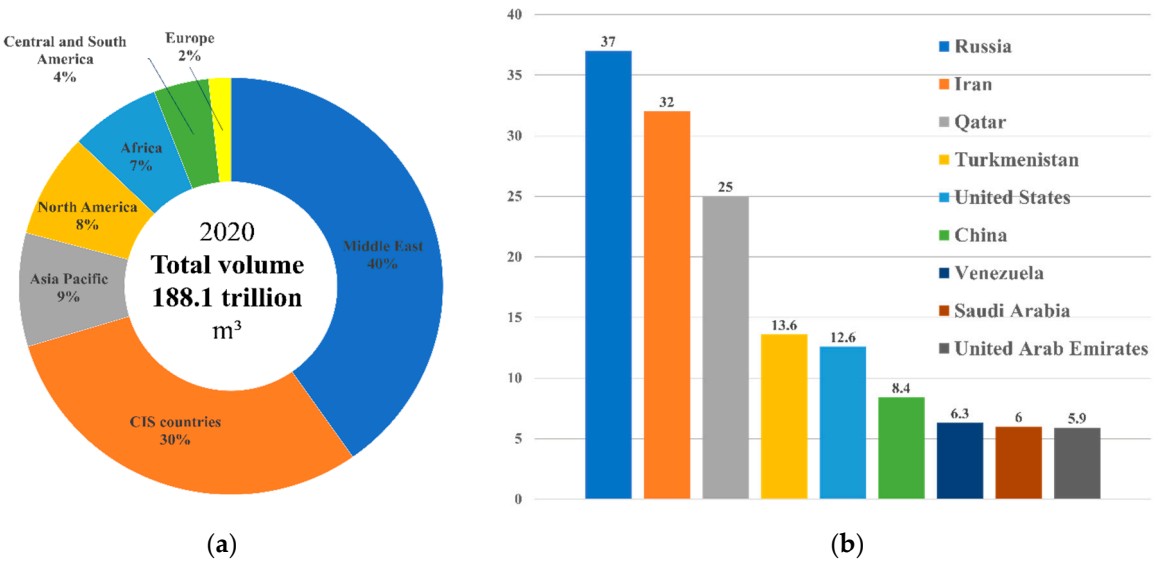

**Figure 1.** (**a**) Distribution of Proven Recoverable Reserves in 2020; (**b**) Ranking of Proven Recoverable Reserves in 2020 ($\times 10^{12}$ m$^3$) (modified from reference [6]).

As an emerging low-carbon energy source, natural gas shows rapid growth in demand. It is expected to become the world's second largest energy source with a demand of $4.62 \times 10^9$ tons oil equivalent in 2040, which is approximately 26% of all the global energy consumption [8]. The development of natural gas in China has also been growing all the time, and the desired amount of natural gas is estimated to be between $5.5 \times 10^{11}$ and $6 \times 10^{11}$ m$^3$ in 2030, which is about 12% of the total energy requirement [9]. Natural gas is the only fossil energy source expected to keep growing under the impact of the COVID-19. IEO indicates that natural gas production shall keep on rising in the coming period to support the growing energy requirement of the developing Asian countries [10].

At ambient conditions, natural gas exists in the gaseous state, and it can be liquefied when the temperature is decreased down to 111.15–114.15 K [11]. Under suitable thermo-dynamic conditions, natural gas could exist as solid gas hydrates [12]. Research related to gas hydrates has been extended in several areas, including desalination [13–15], carbon dioxide capture and sequestration [16,17], gas separation [18–20], refrigeration [21–23], and energy storage and transportation [24–26]. How to efficiently store and transport natural gas is an important issue of natural gas resource utilization. Up to now, people have proposed several kinds of gas storage and transport technologies, which include pipeline natural gas (PNG), liquefied natural gas (LNG), compressed natural gas (CNG), adsorbed natural gas (ANG), and solidified natural gas (SNG). The PNG is a traditional way to transport oil and natural gas on land with high transport volume. About 75% of the world's natural gas is transported by pipeline, while the laying of pipelines is usually restricted by geographical conditions and thus lacks flexibility [27]. Comparatively, the compressed natural gas has greater flexibility and is a more ideal on-board energy source than diesel and gasoline fuels [24]. However, storage and transportation of natural gas at ambient temperature by CNG requires an operating pressure up to 20–25 MPa, which poses potential safety risk of explosion in actual application [28]. In addition, the low storage capacity makes it difficult to use on a large scale [24]. The LNG has a higher storage density than CNG. Nevertheless, the extremely low temperature conditions (111.15 K) required for LNG still results in high storage costs and susceptibility to frost damage in case of gas leakage [11,24,29]. The ANG technique uses adsorbents including carbon nanotubes (CNT), graphene, and metal organic frameworks (MOF) in vessel to adsorb natural gas. Only

3.5–4.0 MPa is required to achieve the goal of natural gas adsorption, while the problem of short life and high cost of the adsorbents have not been solved [30]. Compared to these gas storage technologies, the solidified natural gas is stored in hydrates with solid state, and it is a safe, clean and efficient way to store natural gas [24]. This strategy is a totally new technology for transporting natural gas using hydrates as a carrier, which can be realized through four main steps: the natural gas hydrate formation step, the dewatering step, the granulation step to form hydrate pellets, and the hydrate storage step at certain thermodynamic conditions, as displayed in Figure 2 [24]. However, the SNG technique has not yet been applied on a large scale in industry because of the slow growth rate and the randomness of nucleation during the formation step of gas hydrate. How to promote the synthesis efficiency of gas hydrate, raise the storage capacity and reduce the storage conditions is the key issue for the research of SNG technology [24].

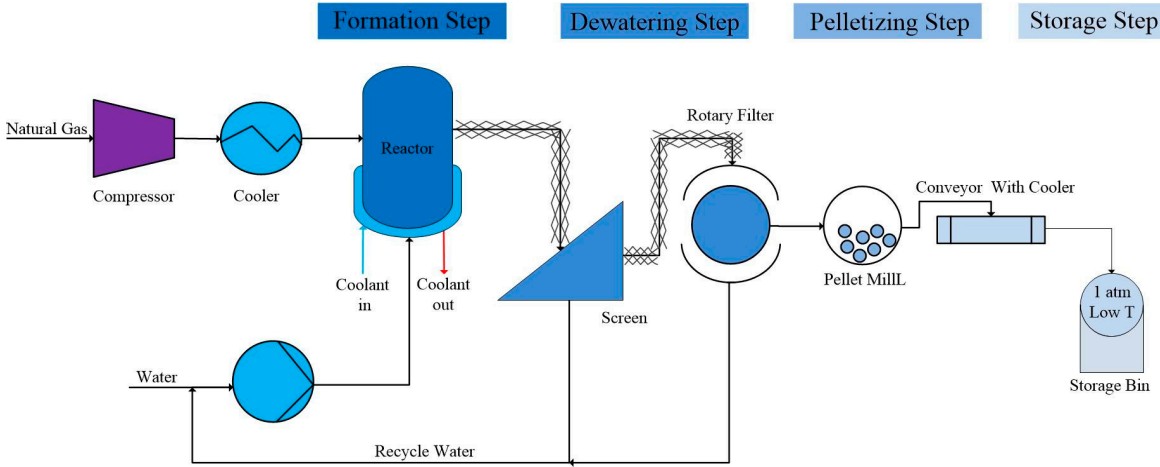

**Figure 2.** The process of natural gas solidification technology [24].

Natural gas hydrate is a cage framework composed of hydrogen-bonded water molecules under high pressure and low temperature conditions [31], and methane gas molecules are associated with water molecules by van der Waals forces to exist within the clathrate to form stable crystalline compounds [32]. Gas hydrates are stable in marine and permafrost conditions [33], which have a wide geographical distribution, and offer significantly bigger reserves than conventional and other unconventional gas resources combined [34]. Natural gas hydrates are a mineral resource that can be mined for energy extraction [31]. The primary premise of extraction is to purposely disrupt the equilibrium temperature and pressure conditions, causing the hydrates to breakdown and release gas [35,36]. Methane gas molecules can be substituted by other kinds of gas molecules (e.g., $CO_2$) in the cages of framework [37].

The formation of gas hydrate is a particularly complicated process, which is often governed by multiple factors [38]. Among the studies on hydrate, researchers have mainly focused on the use of mechanical enhancement methods such as stirring and spraying [39,40], chemical additives [41–46] and other additives [47] to promote natural gas hydrate formation. Hydrate nucleation is often stochastic, which makes the formation process not easy to control, and this problem is more prominent in static reactors [48–50]. According to some research, stirred reactors can effectively reduce the stochastic feature of hydrate during its crystal nucleation and decrease the induction period [51,52]. Stirring could significantly reduce the induction time from about 5 h to approximately 5 min [53]. Under the condition of stirring for 30 min, the speed of hydrate growth rises with increasing stirring rate (320 rmp, 490 rmp, 800 rmp) at 5 MPa [52]. Stirred reactors are effective in reducing the stochastic nature of nucleation compared to stationary systems. For most of the stirred reactor experiments, nucleation is generally achieved within 3 min after the start of stirring [54], which can effectively reduce the hydrate induction time. Therefore, the

majority of experiments are performed in stirred systems. However, the magnetic stirrers may become stuck by the formed hydrate, leading to a large amount of void water and low gas uptake [55]. In addition, blade whole stirring would increase the cost, which is not favorable for industrial applications.

Apart from the mechanical methods, two types of chemical promoters have also been commonly used: the thermodynamic and kinetic promoters. Thermodynamic promoters take effects by shifting the hydrate phase equilibrium curve to more moderate situations (lower pressure and higher temperature), and thus provides larger driving force for hydrate growth. So far, several kinds of thermodynamic promoters have been verified as effective ones, including tetrahydrofuran (THF), cyclopentane (CP), and tetrabutylammonium bromide (TBAB). Among them, tetrahydrofuran (THF) can form sII hydrates (mixed methane and THF hydrates) under more moderate conditions [56]. Studies by some scholars [57–60] showed that in methane systems with THF or CP addition, the large cage was occupied by THF or CP molecules while the small cage provided occupation space for methane molecules. The addition of THF could obviously raise the structure stability of the obtained hydrates when compared to pure methane hydrates [57]. Beheshtimaal et al. [61] confirmed that 5.56 mol% THF was the optimum concentration of the solution to generate hydrate, which could most effectively reduce the pressure while the methane uptake was maximized. Different from the previous two materials, tetrabutylammonium bromide (TBAB) is able to move the hydrate equilibrium conditions to milder zones by forming semi-clathrate compounds with water molecules [62]. Studies have validated that higher stability of hydrates could be achieved by increasing the concentration of TBAB [62,63]. However, as the above thermodynamic promoters usually occupy part of the hydrate cages, the gas storage capacity will be correspondingly reduced, which is also proven in the research by Veluswamy et al. [64].

As another kind of additives, kinetic promoters can drop the hydrate nucleation time and enhance the rate of hydrate generation without causing obvious effect on the equilibrium properties of hydrates. The commonly used kinetic promoters mainly include surfactants [65–67], nanomaterials [68–70], and amino acids [45]. Surfactants (anionic, cationic, and nonionic) are one of the kinetic promoters, such as sodium dodecyl sulfate (SDS). They can enhance the mass transfer occurred at the interface of the gas and liquid by dropping their interfacial tension. Zhang et al. [71] have demonstrated that SDS not only weakens the interfacial tension of the gaseous and liquid phases to raise the solubility of methane, but also has the ability to enlarge the area of the mass transfer interface to accelerate the growth rate of hydrates. Experimental results from Sun et al. [72] also implied that the gas uptake was the greatest in SDS solutions with a concentration of 300 ppm and the anionic surfactant had better facilitation effect than the nonionic surfactant.

Up to now, the reported thermodynamic promoters could participate in the hydration reaction and make it a little faster by changing the formation conditions to milder levels, while their increasing ability is limited. Furthermore, the promoter's occupancy of the cage in the hydrate structure limits the hydrate's gas storage capacity [64]. The surfactant, one of the common kinetic promoters, will produce a large number of foams while the hydrate is dissociated [71,72]. These foams are not conducive to the collection of circulating solutions. It is not only harmful to human body and environment, but also not conducive to cost reduction and energy recovery from the stored hydrate pellets. In contrast to surfactants, only some bubbles in the liquid phase of the hydrate can be noticed by thermal stimulation in amino acid solutions. More importantly, after the hydrate decomposition is completed, no foam remains in the solution of the system, which makes the following treatment procedure much easier for energy recovery [73].

In view of all the additives that have been reported so far, we can see that one main objective is to find more appropriate promoters that have higher efficiency, lower dosage and better environmental friendliness for the successful industrial application of SNG technology. Recently, people have confirmed that some natural amino acids have also the ability of enhancing the formation efficiency of gas hydrate by acting as kinetic promoters. In addition, compared with traditional surfactants, the problem of undesirable bubble

formation can be eliminated by the reported amino acids during the hydrate dissociation stage, and they are also usually accompanied by the environmental-friendly features with relatively low cost. Thus, this article aims to review the mechanism and influence factors of amino acids on the growth behaviors of gas hydrate, including the concentration, the composition of the promoters, and the side chains of these amino acids. A summary related to the thermodynamic and kinetic behaviors of hydrate generation under the effect of various amino acids has been made. Moreover, the influence of amino acids on the morphology of hydrate growth has also been discussed. This paper will provide a thorough overview of the research progress related to amino acids and better guidance for finding more suitable kinetic promoters towards the actual application of SNG technique in industry.

## 2. Effect of Amino Acids Concentration with Different Gas Hydrate Formation Systems

Amino acids are another type of chemicals which show the ability to affect the hydrate formation in recent years. As the basic units of proteins, amino acids are a sort of chemical materials necessary for our society, and they are biodegradable in nature without causing pollution to the environment. Amino acids are usually composed of functional groups, including amino ($-NH_2$), carboxyl ($-COOH$) and side chains [74]. Differences in side chains determine different structures and functions of amino acids. Usually, amino acids may be divided into distinct types based on their side chain groups, as summarized in Table 1 [74,75].

**Table 1.** Classification of amino acids according to the chemical properties and the structures of the side chain groups [74,75].

| Classification of Amino Acids | Aliphatic Amino Acids | Aromatic Amino Acids | Heterocyclic Amino Acids | Heterocyclic Imino Acids |
|---|---|---|---|---|
| Non-polar amino acids (hydrophobic) | Alanine (Ala) Valine (Val) Leucine (Leu) Isoleucine (Ile) Methionine (Met) | Phenylalanine (Phe) Tryptophan (Trp) | / | Proline (Pro) |
| Polar amino acids (hydrophilic) | Glycine (Gly) Serine (Ser) Threonine (Thr) Cysteine (Cys) Asparagine (Asn) Glutamine (Gln) Selenocysteine (Sec) Lysine (Lys) Arginine (Arg) Aspartic acid (Asp) Glutamic acid (Glu) | Tyrosine (Tyr) | Pyrrole lysine (Pyl) Histidine (His) | / |

The deposition of gas hydrates in natural gas and oil pipelines may gradually cause blockage of the pipeline, which further affects the flow of gas and oil, and even poses serious safety accidents. Amino acids were initially investigated as inhibitors of gas hydrates [76–80]. Until 2015, Liu et al. [45] used amino acids as additives with the purpose of accelerating hydrate generation, and verified that some amino acids such as L-leucine could be adopted to be effective promoters to raise the growth speed of methane hydrate. Moreover, they found that not every amino acid was effective in promoting methane hydrate formation. Some amino acids (L−tryptophan, L−methionine, L−arginine, L−phenylalanine, L−histidine, and L−glutamic acid) are favorable for methane hydrate formation. Most of them are lyophobic amino acids. Since then, scholars have started to further investigate the properties of amino acids-promoted hydrate formation for the sake of gas storage. Studies by several authors have shown that higher concentrations of amino acids tend to affect the thermo-

dynamics of hydrates through changing the phase equilibrium for hydrate growth toward lower temperatures and higher pressures, thereby inhibiting hydrate generation [80,81]. Therefore, the concentrations of amino acids employed in the synthesis experiments of hydrates are always low and do not have any effect on the phase equilibrium of gas hydrate. Veluswamy et al. [82] discovered that the absorption rate of the guest gas was accelerated with increasing the concentration of tryptophan, and the kinetics of hydrate formation was enhanced. However, if the concentration of amino acids is raised above a certain level, it tends to increase the mass transfer resistance and further causes an obvious decline in the uptake of the guest gas. In a later study on leucine [73], it was found that methane hydrate in unstirred reactors at leucine concentrations higher than 0.3 wt% would be paste-like and grow rapidly. In contrast, there was no promotion effect when the leucine concentrations were set below 0.3 wt%. Based on the aforementioned results, the promotion ability of leucine at 0.1 wt% concentration was confirmed after combining the operation of the stirred reactor, and the optimal leucine concentration was 0.3 wt% [53]. Prasad et al. [83] found that phenylalanine and methionine were also effective kinetic promoters for gas hydrate synthesis at 0.5 wt% concentration, and that the rates of hydrate generation and gas uptake were both faster with the addition of methionine. Bhattacharjee et al. [84] verified that the growth speed of methane hydrate was obviously raised with the addition of L−histidine, and the enhancement effect increased with increasing L−histidine concentration. The afore-mentioned experimental work showed that amino acids may either promote or inhibit the formation of hydrate. It depends on the amino acid concentration. According to Table 2, it can be found that different amino acids are able to promote hydrate production in the concentration range of 0.3 to 1 wt%.

In addition to pure methane gas, amino acids are also effective promoters in some hybrid systems containing mixture gases. Veluswamy et al. [85] conducted a study of hydrate formation with mixed natural gas (methane-ethane-propane). They found that only 0.02 wt% of tryptophan presented the feasibility of promoting the hydrate growth with mixed natural gas. Jeenmuang et al. [86] found that the optimal concentration of valine to promote methane uptake was 0.25 wt%, and the optimal concentration of Leucine and methionine was 0.125 wt%. Li et al. [87] proved that when the concentration of valine was less than 0.077 mol%, there was no remarkable acceleration effect on the generation of HCFC-141b hydrate. The addition of a small amount of amino acid could only disperse a small portion of HCFC-141b in water, and the chance of hydrate nucleation was less, and the induction time was longer when compared with higher concentrations. On the other hand, excessive amount of amino acid makes the reaction solution easy to be stratified, which is also not conducive to the hydrate formation. This work means that amino acids have a strong applicability. It is not only limited to methane hydrate formation but can also be widely applied to more conditions.

In the preceding work, methane hydrate formation and storage are performed under laboratory situations, which are often associated with high pressure and low temperature. However, for commercial manufacturing, not only are the formation conditions difficult to achieve, but also the energy consumption and production costs are significant. Another study by Bhattacharjee et al. [88] showed that the combination of 1,3-dioxolane (DIOX) with 300 ppm tryptophan enables fast speed of hydrate generation, and the stability of the formed hydrates can be achieved at the temperature of 268.15 K under atmospheric pressure. This study demonstrates the potential of suitable concentration of amino acid for future commercial use of SNG technology.

Seawater is usually thought to be an accessible raw material for hydrate formation in industry. One of the salt materials contained in seawater is the sodium chloride (NaCl), which can perform as a thermodynamic inhibitor to prevent hydrate growth by changing the equilibrium profile of hydrates to higher pressure and lower temperature [89]. The increase of NaCl concentration enhances the inhibition of hydrate formation. Low NaCl concentration of 0.2 mol/L helped to decrease the hydrate induction time during its formation [90]. However, a study by Ren et al. [91] showed that 50 mmol/L low concentration of NaCl and

anionic surfactant could be still favorable for hydrate formation. Veluswamy et al. [92] also found that 0.02 wt% Leucine (hydrophobic amino acid) could improve the growth efficiency of mixed hydrate (methane and THF) when the saline and seawater were present in the system. Bhattacharjee et al. [93] conducted the formation experiments of methane hydrate in seawater using different combinations of 0.03 wt% amino acids (arginine and tryptophan) with THF and TBAF. The results showed that amino acids with other additives were able to improve the formation kinetics of hydrate to some extent under their synergistic effects. Inkong et al. [94] employed several mixed promoters which were composed of SDS and amino acids such as methionine, Leucine, and valine) to investigate the hydrate growth behaviors in brine in a room temperature (298.2 K) environment at concentration of 0.05 wt%. The results showed that the existence of SDS and amino acids increased the speed of mixed hydrate formation in brine species and improved the kinetics of hydrate generation. Veluswamy et al. [95] confirmed that the acceleration ability of Leucine was outstanding when it was adopted for the formation of mixed natural gas (methane-ethane-propane) hydrate at the concentrations of 0.01 and 0.02 wt% in brine. When seawater is selected as the experimental material, some amino acids can work together with other promoters and can reduce the concentration of amino acids, according to Table 2.

Based on studies by different scholars, Nasir et al. [96] concluded that the amino acid concentration equal or greater than 5 wt% had a relatively inhibitory effect, and no thermodynamic inhibition was observed in low concentrations (1 wt%) of amino acid. The interactions of amino acids with water molecules are essential in the creation of natural gas hydrates. Using a higher concentration in a gas hydrate system can attract more water molecules to inhibit hydrate formation.

In general, several kinds of amino acids have been confirmed to be able to take effect in accelerating the crystallization speed of natural gas hydrate, and the promotion effect increases with the rise of concentration in a certain range. In addition, they can be combined with some other chemical substances such as SDS, THF, and DIOX to further change the hydrate equilibrium to more moderate temperature and pressure conditions. Such combination is effective in both pure water and seawater systems with methane or mixed natural gases. The typical concentrations at which various amino acids can facilitate the hydrate formation in different systems have been summarized in Table 2.

**Table 2.** Concentration of amino acid-promoted hydrate formation in different systems.

| Added Amino Acids | Reaction System | P/MPa | T/K | Concentration/wt% | Ref. |
|---|---|---|---|---|---|
| Tryptophan | $CH_4$ | 10 | 275.2 | 0.3 | [82] |
| Histidine | $CH_4$ | 10 | 275.2 | 1 | [82,84] |
| Arginine | $CH_4$ | 10 | 275.2 | 1 | [82] |
| Leucine | $CH_4$ | 10 | 275 | 0.5 | [73] |
| Leucine | $CH_4$ | 10 | 275.2 | 1 | [53] |
| Methionine | $CH_4$ | 5.3 | 275 | 0.5 | [83] |
| Phenylalanine | $CH_4$ | 5.3 | 275 | 0.5 | [83] |
| Tryptophan | Mixed natural gas | 5 | 283.2 | 0.03 | [53] |
| Valine | $CH_4$ + 5.56 mol% THF | 8 | 293.2 | 0.25 | [86] |
| Leucine | $CH_4$ + 5.56 mol% THF | 8 | 293.2 | 0.125 | [86] |
| Methionine | $CH_4$ + 5.56 mol% THF | 8 | 293.2 | 0.125 | [86] |
| Arginine | FW + $CH_4$ + 5.56 mol% THF | 9.5 | 298.2 | 0.05 | [54] |
| Leucine | Salt water + $CH_4$ + THF | 5 | 283.2 | 0.03 | [92] |
| Arginine | Salt water + $CH_4$ + THF | 5 | 283.2 | 0.01 | [92] |
| Arginine | Salt water + $CH_4$ + 5.56 mol %THF | 9.5 | 298.2 | 0.03 | [93] |
| Tryptophan | Salt water+ $CH_4$ +5.56 mol %THF | 9.5 | 298.2 | 0.03 | [93] |
| Valine | Salt water+ $CH_4$ +5.56 mol %THF | 8 | 288.2 | 0.05 | [94] |
| Leucine | Salt water+ $CH_4$ +5.56 mol %THF | 8 | 288.2 | 0.05 | [94] |
| Methionine | Salt water+ $CH_4$ +5.56 mol %THF | 8 | 288.2 | 0.05 | [94] |
| Tryptophan | Salt water + Mixed natural gas | 7.2 | 283.2 | 0.02 | [95] |
| Leucine | Salt water + Mixed natural gas | 7.2 | 283.2 | 0.02 | [95] |

### 3. Effect of Amino Acids Side Chains on Hydrate Formation

The study by Bavoh et al. [97] found that arginine and valine, which had long side chains, could promote hydrate kinetics. One possible reason was that the arginine and valine might be able to hinder the aggregation of hydrate particles at the gas-liquid contacting surface during hydrate crystallization. Such behavior provides more favorable conditions for the diffusion of methane molecules into the liquid water and therefore increases the gas storage capacity during hydrate generation [98]. The side chain length of amino acids is generally considered to be a critical factor affecting the kinetics of hydrate formation [82]. Nasir et al. [96] have mentioned that valine showed better inhibitory efficiency than arginine because of its shorter chain length and better solubility in water. Short side chains tend to interact with water to form hydrogen bonds, thus preventing water molecules from forming clathrate with gas and inhibiting hydrate formation [82]. Conversely, longer or larger side chains resist the occurrence of hydrogen bonds with water, and thus promote the kinetics of hydrate formation [82]. Histidine and arginine used in the study of Veluswamy et al. [82] moderately promoted methane hydrate formation, which was attributed to their longer side chains. In the formation experiments of carbon dioxide hydrate, Cai et al. [99] similarly discovered that the side chain length of amino acid was responsible for the enhanced hydrate kinetics. Their study showed that Leucine and methionine had the optimal side chain lengths to increase the rate of carbon dioxide hydrate formation while increasing methane uptake.

Another class of studies suggests that the nature of the side chains of the amino acids poses an outstanding effect on hydrate growth. When the hydrophobicity of the side chain is weak, it tends to inhibit the hydrate formation [100]. Conversely, when the side chain is more hydrophobic, it promotes hydrate formation better [99]. Bavoh et al. [97] attributed the higher gas hydrate acceleration ability of valine than arginine to the hydrophobicity of its side chain. Some other authors [87] also found that as the hydrophobicity of amino acids increased, the required dosage of amino acids that were most effective in promoting hydrate nucleation gradually decreased. Jeenmuang et al. [86] showed that the rate of hydrate generation under the combination effect of amino acids with hydrophobic side chain (similar to methionine, Leucine and valine) and the thermodynamic promoter tetrahydrofuran (THF) was 5 times as high as that obtained using the THF solution alone. Bhattacharjee et al. [54] showed that tryptophan (hydrophobic amino acid) promoted mixed (methane and THF) hydrates better than arginine (hydrophilic amino acid) at room temperature.

Later, Veluswamy et al. [82] compared the formation characteristics of methane hydrate in three different systems containing histidine, arginine, and tryptophan, respectively. It was found that tryptophan had the most remarkable enhancing effect on hydrate formation, inferring that the hydrophobic nature and the existence of aromatic side chains could better promote methane hydrate formation. Prasad et al. [83] employed two kinds of hydrophobic amino acids with different side chains, and discovered that the methane hydrate generation rate was faster for phenylalanine (aromatic group) and the methane gas uptake was slightly higher than that of methionine (aliphatic group). These scholars concluded that aromatic amino acids promoted hydrate formation more effectively than aliphatic.

In summary, people have verified that low concentration of several amino acids can promote hydrate formation. These amino acids are generally hydrophobic and have long side chains. Similar to surfactants, hydrophobic amino acids can reduce the tension at the gas-liquid contacting surface and thus improve the mass transfer between the gas and water phases. The short side chains are prone to interact with water to form hydrogen bonds, thus preventing water molecules from forming a cage-like framework linked by hydrogen bonds and inhibiting the formation of hydrates. Conversely, longer side chains are not favorable for the occurrence of hydrogen bonds with water and can promote the kinetic formation of hydrates. In other words, the selection of amino acids should be focused on the categories which have hydrophobic and relatively long side chains. In addition, some people consider that aromatic amino acids have better promotion effect than aliphatic.

## 4. Thermodynamic and Kinetic Behaviors of Hydrate Formation with Amino Acids

### 4.1. Effect of Amino Acids on the Thermodynamic Features of Hydrates

Experiments performed on methane hydrate formation revealed that some amino acids act as thermodynamic inhibitors [79,101]. Bavoh et al. [102] studied and compared four amino acids (threonine, valine, asparagine and phenylalanine) and found that the inhibitory effect of 5 wt% valine was the most obvious, followed by threonine, then asparagine and phenylalanine. Moreover, valine and threonine were found to have the smallest molecular weights and their inhibition performance on methane hydrate growth is more significant compared with phenylalanine and asparagine. Furthermore, the base acid concentration at 10 wt% showed the best inhibition effect. The suppression abilities of 10 wt% amino acids can be displayed in the following descending sequence: glycine > alanine > proline > serine > arginine [80]. Scholars believe that the above order of inhibitory effects of amino acids may depend on their molecular weights, with the higher molecular weight amino acids having higher inhibitory effects in most cases. The experimental results of Bavoh et al. [97] showed that there was an equilibrium temperature reduction of approximately 0.5 K on average for gas hydrate with the existence of arginine and valine, which thermodynamically exhibited inhibition of methane hydrate formation. The exhibited restraining effect of arginine and valine may be because they are able to construct stable hydrogen bonds with water molecules [81]. In addition, the inhibition ability of alanine and serine was found to be stronger than arginine, which was because the latter had longer side-group alkyl chains.

According to Figure 3a [102], amino acids hardly affect the thermodynamics of the hydrate at 1 wt% concentration when compared with pure water phase equilibrium curve [103]. At the 5 wt% and 10 wt% concentration of amino acid, the phase equilibrium curve shifts slightly toward high pressure and low temperature (Figure 3b,c) [80,97,102,104]. In addition, the inhibition capacity increases with the increase of amino acid concentration, as shown in Figure 3d [80]. The formation conditions of methane hydrates have been shown to become more severe under the effect of amino acids (high concentration) in almost all thermodynamic studies. The existence of amino acids at this time changes the methane hydrate phase equilibrium profile to the region with higher pressure and lower temperature. Under the same concentration, Alanine, Glycine, Lysine and Proline showed better inhibition efficiency than Serine and Arginine because of their shorter chain length and better solubility in water, as shown in Figure 3c [80,104]. The inhibitory performance of these amino acids is caused by their hydrogen bonds formed with water molecules, which drops the ability of water in generating gas hydrates [79,105,106]. The variation in the strength of methane hydrate inhibition among different amino acids is caused by the differences in the physical and chemical properties of their side chains [80,81]. The length and properties of the amino acid side chain are also associated with their molecular weights. Amino acids with smaller molecular mass are often accompanied with shorter chain length and stronger hydrophilic features, which results in more intense hydrogen bonding interactions with water and thus more serious restraining effects on hydrate formation.

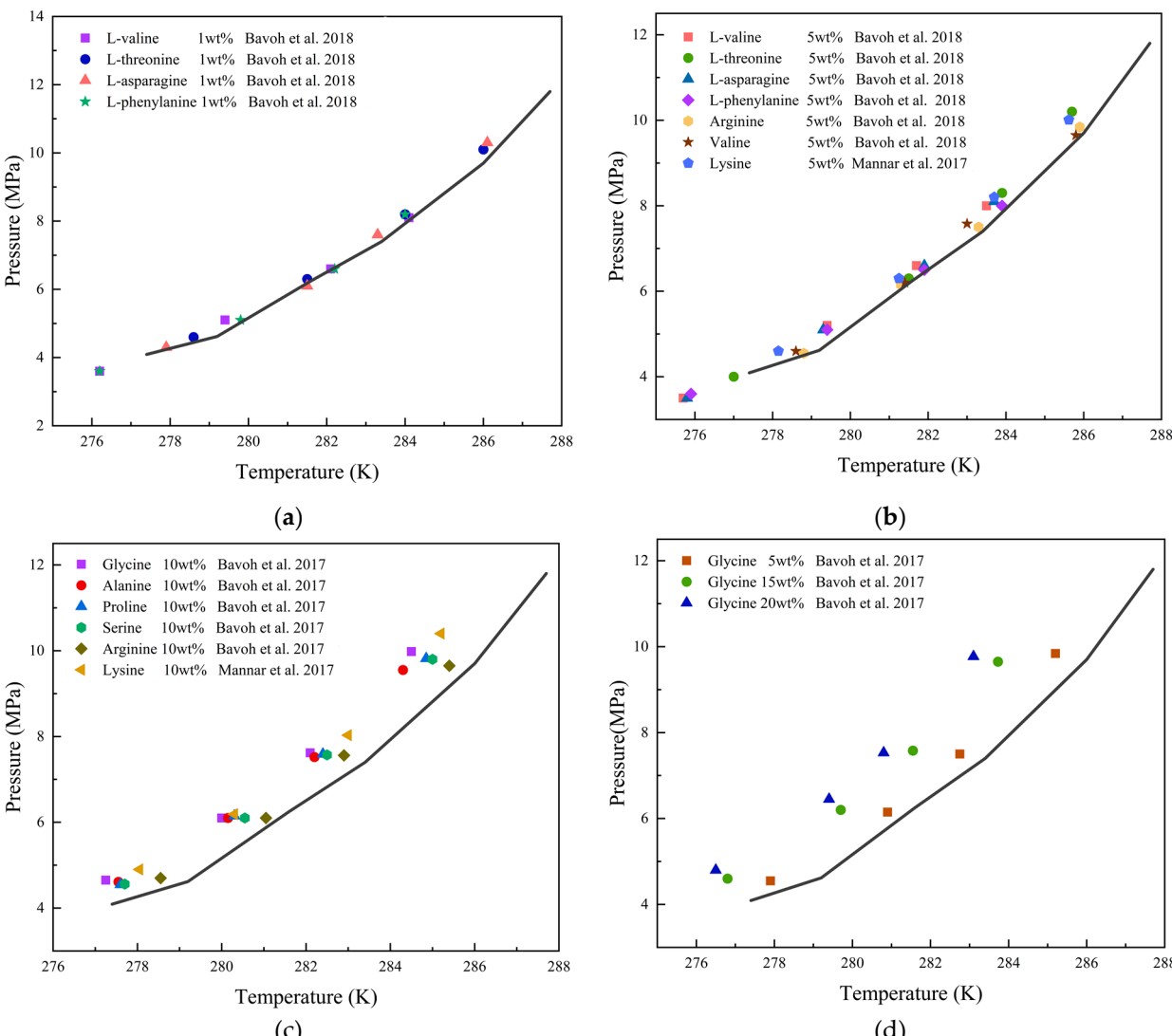

**Figure 3.** (**a**) Phase equilibrium curve of gas hydrate under the action of 1wt% amino acids [102]; (**b**) Phase equilibrium curve of gas hydrate under the action of 5wt% amino acids [97,102,104]; (**c**) Phase equilibrium curve of gas hydrate under the action of 10wt% amino acids [80,104]; (**d**) Phase equilibrium curve of gas hydrate under the action of different concentrations of Glycine [80] (The solid line indicates the pure water phase equilibrium curve, modified from reference [103]).

### 4.2. Effect of Amino Acids on the Kinetics of Hydrate Formation

The effect of amino acids on the kinetics of gas hydrate formation has been another focus of all the researchers. Bavoh et al. [97] found that the induction time of gas hydrate in valine solution was longer than that of pure water system at 7.1 MPa and 274.15 K. However, the arginine-affected induction time was nearly the same with that of water, as presented in Figure 4a. Li et al. showed [87] that the amount of the added amino acids affects the induction time of hydrate formation. As the concentration of amino acids increased, the induction time firstly decreased and then increased. The shortest average induction time of hydrate generation was obtained when the concentration of the added amino acids was about 0.200 mol%. Bhattacharjee et al. [54] showed that tryptophan (hydrophobic amino acid) at a concentration of 500 ppm promoted mixed (methane and THF) hydrates better than arginine (hydrophilic amino acid) at room temperature at 3.1 MPa and 298.2 K. Figure 4b [53,82] presents the variation of hydrate induction time with the addition of different types and concentrations of amino acids at 275.2 K and 10 MPa. The randomness

of hydrate nucleation could be reduced while an average induction time of no more than 3 min could be observed for all studied systems. However, because of the stochastic feature of nucleation, the variation in induction time is not necessarily reduced under the same experimental conditions. As shown in Figure 4b [53,82], only the induction time decreased after the addition of Histidine. With increasing amino acid concentration, the induction time of hydrate with the addition of Tryptophan was even much longer than that of pure water. Due to the stochastic nature of hydrate nucleation, the use of induction time alone is sometimes misleading. Therefore, hydrate formation rate versus total methane moles consumed was used to further research the effect of amino acids on methane hydrate formation.

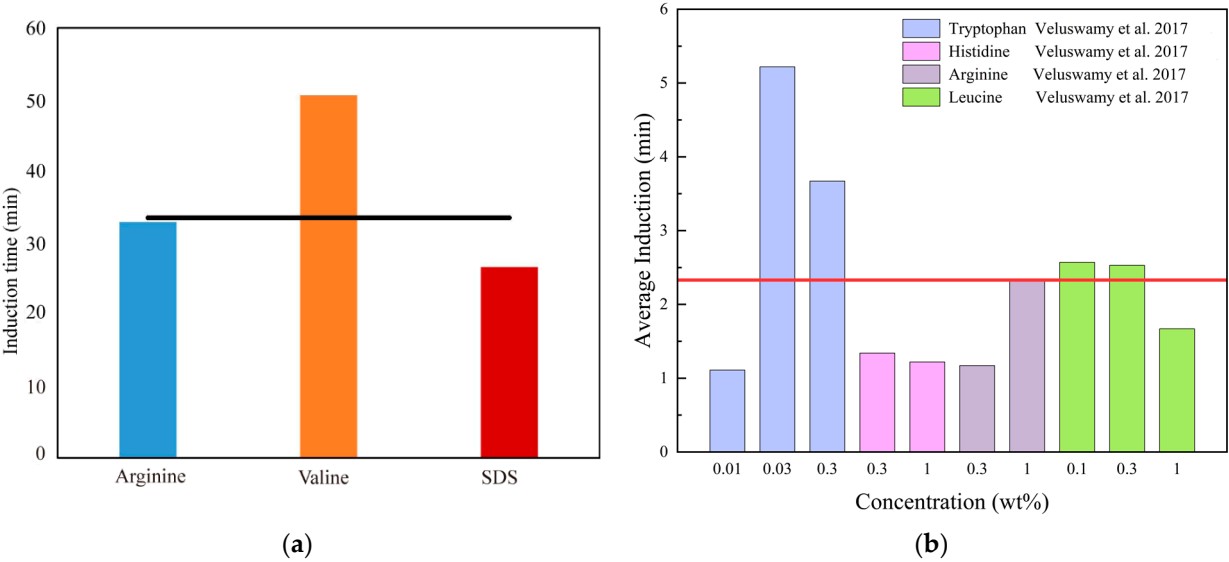

**Figure 4.** (**a**) The induction time of hydrate formation under the impact of 1wt% arginine, valine and SDS at 7.1 MPa and 274.15 K (the induction time in pure water system is represented by the solid line) [97]; (**b**) Comparison of induction time of hydrate under the action of amino acids at 10 MPa and 275.2 K (the red line stands for pure water system) (modified from reference [53,82]).

The study of Veluswamy et al. [73] observed a deflection point in the system with amino acid (Leucine) at 10 MPa and 275 K. It was observed that the rate of hydrate growth increased significantly after the deflection point. It is attributed to the fact that the gas uptake always lags behind the disappearance of liquid water and the formation of solid hydrates. Another study by them showed that the speed of hydrate generation could be raised when the Leucine concentration was increased to 1 wt% at 275.2 K and 10 MPa [53]. Bhattacharjee et al. [84] found that the average hydrate crystallization rate in the solution with 1 wt% L−histidine was almost three times as high as that in pure water system at 274.15 K and 5.0 MPa. For the mixed gas system containing methane, ethane, and propane the rate of hydrate formation was measured to increase with increased amino acid concentration at 283.2 K and 5.0 MPa [85]. Bavoh et al. [97] confirmed that the addition of arginine and valine dropped the rate of hydrate formation to a lower level than that in pure water at 7.1 MPa and 274.15 K, especially in valine system where hydrate formation was the slowest, as shown in Figure 5a. Among them valine shows the highest reduction in the growth speed of methane hydrate. Burla et al. [107] discovered that gas uptake rate was quicker in the L−methionine and L−phenylalanine systems (t90 at about 40 min), and sluggish reaction kinetics was seen in the L−cysteine, L−valine, and L−threonine systems at 5.5 MPa and 298 K. Figure 5 presents the comparison of formation rates with different amino acids at a concentration of 1 wt% at 275.2 K and an initial pressure of 10 MPa. According to Figure 5, it can be observed that arginine, valine and histidine all tended to make the rate of hydrate formation decline. Among them valine shows the

highest reduction in the growth speed of methane hydrate. The promotion of hydrate generation rate by leucine is the most significant. Among these three amino acids, only leucine is a hydrophobic amino acid.

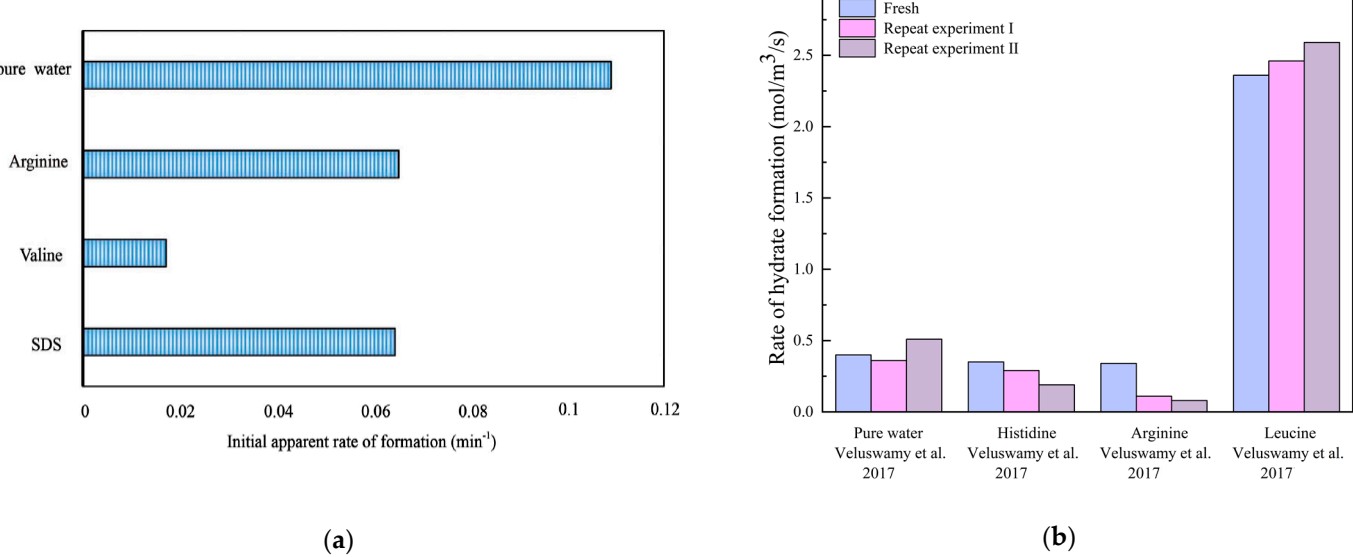

(**a**) (**b**)

**Figure 5.** (**a**) The early rate of methane hydrate formation under 1wt% arginine, valine, and SDS at 7.1 MPa and 274.15 K [97]; (**b**) The rate of hydrate formation under the 1 wt% of histidine, arginine and leucine at 10 MPa and 275.2 K (Three sets of columns with different colors represent the results of three sets of repeated experiments) (modified from reference [53,82]).

The gas uptake is another key parameter correlated with the kinetics of hydrate formation. Veluswamy et al. [73] found the highest gas uptake at 0.3 wt% Leucine concentration at 10 MPa and 275 K, but the gas uptake decreased when the Leucine concentration increased to 1 wt%. Bhattacharjee et al. [84] found that the methane consumption of 1 wt% L−histidine was comparable to the gas uptake of 1 wt% SDS at 274.15 K and 5.0 MPa. Gas uptake was not proportional to concentration, because 300 ppm tryptophan could achieve a faster hydrate formation rate, while the gas uptake was about 10% lower than that observed in experiments with pure water and with tryptophan solutions at lower concentrations at 283.2 K and 5.0 MPa [85]. Bavoh et al. [97] found that the methane gas uptake in the using of arginine and valine was about 3 and 10 times as high as that of pure water system at 7.1 MPa and 274.15 K, respectively. The methane gas uptake with valine was 1.5 times higher than that of SDS, while arginine did not differ much from SDS. Gaikwad et al. [108] found that the addition of 1 wt% tryptophan and cyclooctane significantly increased the rate of hydrate formation and gas uptake compared to the mono cyclooctane system at 274 K and 5.0 MPa. A study by Burla et al. [107] demonstrated that the several amino acids used (L−methionine, L−phenylalanine, L−cysteine, L−valine, and L−threonine) had 80–85% gas absorption capability, with the exception of L−threonine, which had just 30 percent overall uptake capacity. At concentrations below 1 wt%, histidine, arginine, Leucine and tryptophan were able to promote gas uptake, as shown in Figure 6a [53,82]. It is worth mentioning that arginine and valine are less effective in influencing the induction time and initial formation rate, but they both promote gas uptake greater than SDS, as shown in Figure 6b [97]. Bavoh et al. [97] think that the promotion of arginine and valine can be attributed to the fact that they both have longer alkyl side chains [77]. In addition, the promotion effect of valine is better than that of arginine, which the authors attribute to the fact that the methyl R group in valine can be incorporated as a guest in the hydrate cage [109]. This can provide some stability to hydrate formation and therefore lead to higher hydrate formation than arginine. According to Figure 6c [53], when the concentration of amino acid increases, the rate of hydrate formation accelerates. However, the gas

uptake decreases. It is possible that the large amount of hydrate formation in the early stages hinders the later reactions.

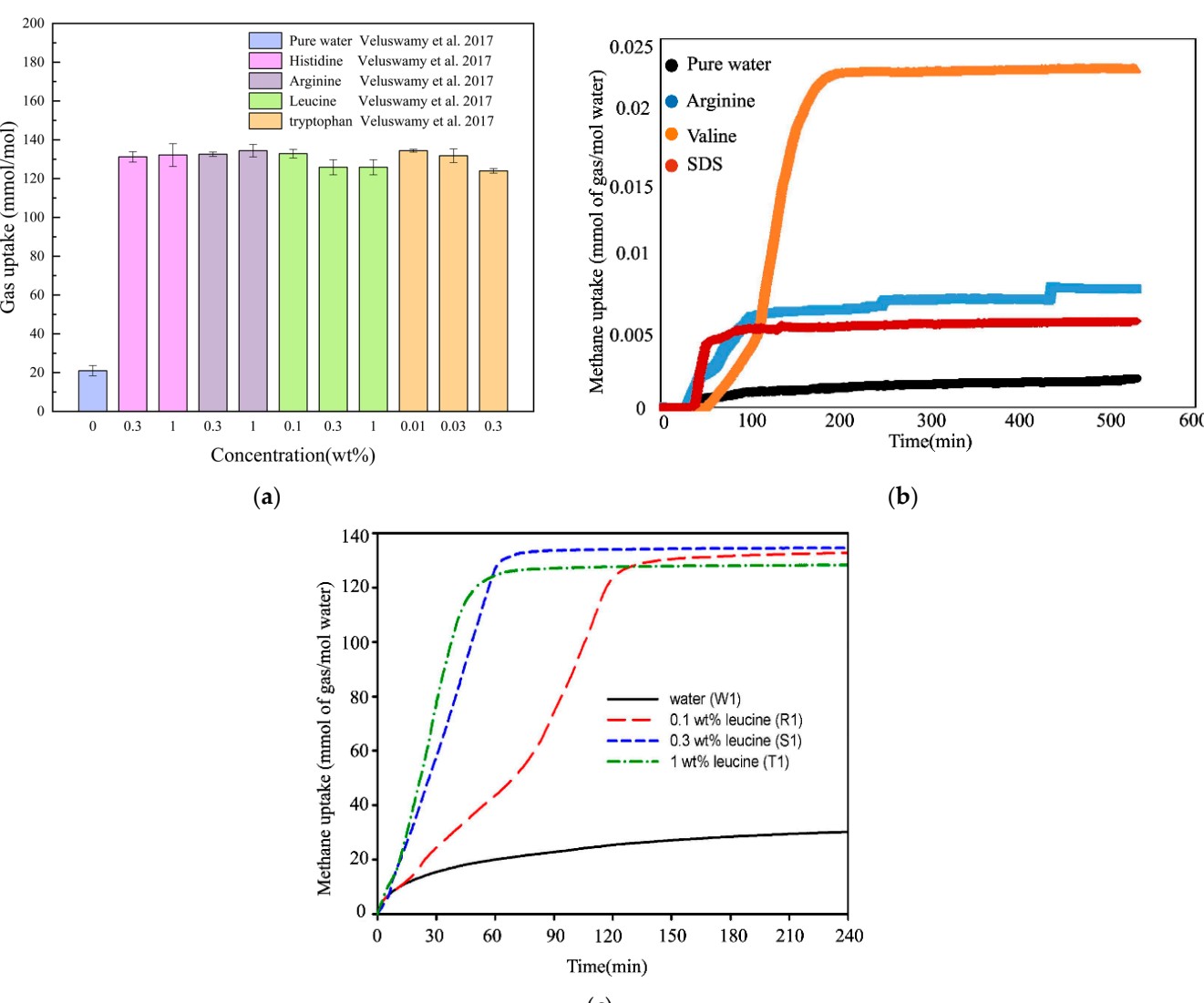

(**a**)

(**b**)

(**c**)

**Figure 6.** (**a**) Comparison of gas uptake under different concentration of amino acids (modified from reference at 10 MPa and 275.2 K [53,82]); (**b**) Gas uptake under 1wt% of amino acids at 7.1 MPa and 274.15 K [97]; (**c**) Comparison of gas uptake by using different concentrations of leucine at 10 MPa and 275.2 K [53].

In summary, contrast to pure water systems, amino acids are active in reducing the induction time of hydrates, while the induction time is not stable because of the random nature of hydrate nucleation. Amino acids have a beneficial influence on the kinetics of hydrate formation at specified concentrations. In general, the influence of amino acids on hydration increases as concentration rises. That is, the rate of hydrate formation and gas uptake were both effectively increased, while the relationship between the promotion ability of amino acids and the concentration was not always positive. When the concentration of amino acids improves to certain amount, the rate of hydrate formation can be still effectively increased, while the expense is that the gas storage capacity will be obviously reduced. This might be due to the high concentration of amino acids, which causes the hydrate formation rate to be excessively quick. and the generated hydrate will accumulate in large quantities on the gas-liquid contact surface to form a thick hydrate layer, which instead increases the

mass transfer resistance and hinders the gas absorption ability of the residual water in the later formation stage. Therefore, the concentration of amino acids should be determined cautiously by simultaneously considering its positive effect on hydrate growth rate and negative effect on gas storage capacity.

### 4.3. Effect of Amino Acids on the Morphology of Hydrate Formation

The study of hydrate formation morphology can be achieved by observing the shape, size, and growth direction of hydrates during its formation [110]. The hydrate formation morphology can help to complement the study of hydrate formation kinetics including the analysis of the causes of changes in formation rates, gas storage volumes and other factors, and can further improve the investigation of hydrate formation mechanisms. It will help to design reactors with faster formation rate and higher gas storage capacity, and provide theoretical basis for future industrial application. The nucleation of gas hydrates takes place at the gas-liquid contacting interface, because the gas is primarily dissolved at the gas-liquid interface. As hydrates continue to be generated and accumulate, the hydrate film becomes more and more thick, and it eventually forms a dense hydrate layer. The dense layer shall prevent the gas from dissolving in the solution, so that no more hydrate can be generated in the liquid phase or in the gas phase at the later stage of the formation reaction. The morphology of the gas hydrate formed in pure water system is shown in Figure 7 [82].

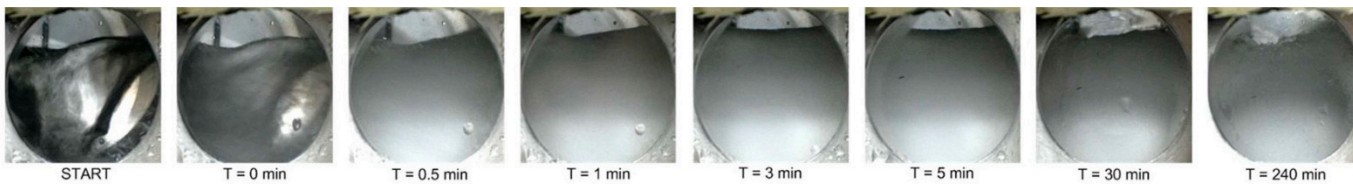

**Figure 7.** Shape of hydrate formation in pure water at 10 MPa and 275.2 K [82].

However, the growth pattern of hydrates changes significantly after the addition of amino acids. The most notable disparity is that the hydrate crystals grow upward in the gas phase or downward in the liquid phase (Figure 8 [73]). Veluswamy et al. [73] firstly observed the morphology of methane hydrate formation after addition leucine. The results showed that in the presence of amino acids, the morphology of methane hydrate formation changed obviously, and the formed hydrate was able to be dissociated completely without foam formation. They observed that the nucleation and growth of hydrate still occurred at the gas-liquid contact interface after the addition of Leucine. After nucleation, the hydrate started to climb up the reactor's wall, forming a pit-like structure. At this time the hydrate growth in the liquid phase was relatively less prominent. However, as the reaction proceeds, hydrate formation in the liquid phase grows rapidly. Subsequently, accelerated growth of hydrates in both the gas and liquid phases can be observed. After the hydrate covers the whole liquid phase, the solid crystals will continue to expand in the gas phase. After the hydrate occupies the whole observation surface, the hydrate keeps thickening, hardening and forming vein-like structures. Finally, the hard hydrate covers the whole observation surface and no further morphological observation can be performed. When the morphology of the hydrate is further magnified, capillary-like channel structures can be observed. The hydrate with amino acids is porous in nature, and the dense hydrate generated by pure water. These channels can enhance the touch between the gas phase and the liquid phase, further promoting the hydrate growth. In addition, special bubble-like connecting channels existing between the hydrate layer and the liquid phase can be observed in leucine solution with the concentration of 0.3 wt%. A translucent layer can be observed near the channels connecting the bubbles to the hydrate layer. Over time, the bubble gradually inflated and hydrate on the surface gradually grows. After that the translucent hydrate layer transforms into a dense hydrate layer. The authors infer that the bubble channel helps the gas above to pass through the hydrate layer into the liquid below and thus promotes the hydrate growth.

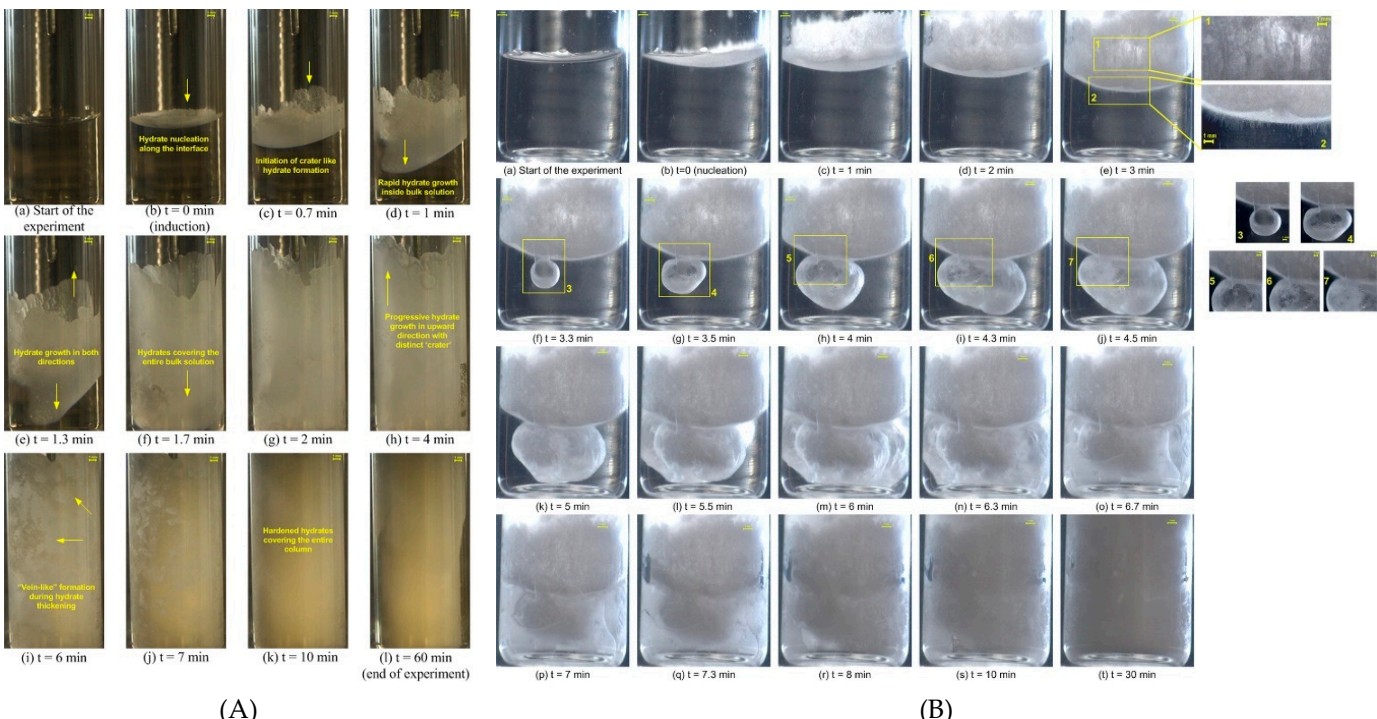

**Figure 8.** (**A**) Formation of hydrate in amino acid solution; (**B**) Shape of hydrate formation in 0.3 wt% leucine solution at 10 MPa and 275 K [73].

In summary, the process of hydrate formation in the addition of amino acids is obviously differs significantly from that in pure water. Similar to pure water, the nucleation of hydrate also occurs at the gas-liquid contact interface. The main difference is that hydrate formation in pure water occurs only at the gas-liquid contact interface, while gas hydrates grow along the glass tube wall into both gas and liquid phases when the amino acids are present. Moreover, in contrast with the dense hydrate generated in pure water, the hydrate formed in the amino acid solution has a large number of pores and capillary channels. The morphological analysis suggests that these porous, pit-like structures of hydrates allow more liquid to be drawn through these pores and channels to the gas-liquid contact interface under the capillary suction, which is equivalent to increasing the contacting chances between the gas and liquid phases and thus speeding up the hydrate growth kinetics.

### 4.4. Influencing Mechanism of Amino Acids on Hydrate Formation

The mechanism of low concentrations of amino acids to promote the kinetics of methane hydrate formation can be hypothesized to have similarities with surfactants based on the existing scholar studies [45,82]. Microscopically, the unique amphiphilic nature of amino acid molecules allows the hydrophilic groups to be attracted and soluble in water, while the hydrophobic sections tend to be repelled by water. Amino acid hydrophobic groups can adsorb at the hydrate interface, lowering the surface tension between the two phases and decreasing the hydrate's nucleation difficulty [111]. In addition, if the side chains of amino acids are long enough, they will resist the formation of hydrogen bonds with water and can regulate the kinetics of hydrate formation positively [82,99]. It can also disturb the water structure around the hydrate, prevent the aggregation of hydrate particles, strengthen the local water structure, and increase the contact surface between water and gas [87].

Macroscopically, gas hydrate grows along the glass tube wall into both gas and liquid phases after adding amino acids. The hydrates generated in solution are usually accompanied by a huge number of pores that resemble capillary channels. These porous, pit-like structures of hydrates allow more liquid to be drawn through these pores to

the gas-liquid contact interface by capillary effect, which corresponds to an increase in the contact channels between the gas and liquid phases, thus speeding up the hydrate generation kinetics [73]. Based on the preceding work, we presume that amino acids with low concentrations can facilitate hydrate formation, and as the concentration increases, the rate of gas uptake is accelerated and the kinetics of hydrate formation is enhanced. However, this enhancement effect is not without limitation. When the amino acid concentration reaches a specific amount, the generated hydrate will accumulate heavily on the gas-liquid contact surface and form a thick hydrate layer, which will in turn raise the mass transfer resistance and hinder further hydrate formation. Therefore, the amino acid boosting impact on hydrate may be attained within a certain concentration range. Beyond a specific range, perhaps it will show an inhibitory effect. Compared with other aspects of hydrate research, the study of amino acid promoted hydrate formation is relatively late and less conducted, and the promotion mechanism of amino acid on hydrate formation still needs further study.

## 5. Conclusions and Outlook

The SNG technology is a promising, efficient, safe and environmental-friendly strategy of natural gas storage in the form of solid gas hydrates. This study provides an overview of the investigations of methane hydrate production undertaken by employing various amino acids for the purpose of natural gas storage. Based on the analyses of the effect of the amino acids on the formation properties of gas hydrate, the following conclusions can be obtained:

(1) Amino acids with high concentrations generally show an inhibitory impact in almost all the thermodynamic tests, while low quantities have no effect. The methane hydrate phase equilibrium curve changes toward the high pressure and low temperature area when high amino acid concentrations are present. At specified concentrations, amino acids have a favorable influence on hydrate formation kinetics, but too high concentrations enhance the pace of hydrate formation at the expense of gas uptake. This might be because high amino acid concentrations cause hydrate formation to occur too quickly, causing the created hydrate to gather heavily on the gas-liquid contact surface, which generates a thicker hydrate layer that increases mass transfer resistance and prevents hydrate formation. As a result, the choice of amino acid is critical.

(2) Low concentrations of amino acids are often favorable for kinetic promotions of hydrate formation, especially when used in combination with other promoters. Future research can be expanded with a wider variety of combined promoters to explore lower addition of additives, faster hydrate formation rates, and higher gas uptake. Stirring in laboratory reactors is effective in reducing the stochasticity of hydrate nucleation, but stirring throughout the whole hydrate production process is not conducive to cost savings. Therefore, the exploration of new semi-stirred reactor systems is an important part of SNG commercialization.

(3) In the presence of amino acids, the morphology of hydrate formation is similar to that of pure water, while yet differing. The nucleation and first formation of hydrates occur at the gas-liquid contact interface, which is similar to pure water. The distinction is that hydrate formation in pure water occurs exclusively at the gas-liquid contact interface, whereas hydrates develop in pore form along the glass tube walls into both gas and liquid phases when amino acids are present. In this paper, the referred morphological study is only conducted at the macroscopic level, and further analysis needs to be performed at the microscopic scale to clarify the mechanism of the contribution of amino acids to the change of hydrate structure.

(4) Low concentration of amino acids can promote hydrate formation because hydrophobic groups can be adsorbed at the hydrate interface, which reduces the interfacial tension between the two phases. In addition, amino acids with long side chains are less likely to form hydrogen bonds with water, and they do not hinder water molecules from clathrate with gas molecules, which can speed up hydrate formation kinetics.

The formed hydrates are frequently porous in character in the presence of amino acids, increasing the interaction between the liquid and gas phases through these porous channels under the action of capillary suction. However, further research is needed to fully explain the mechanism by which amino acids stimulate hydrate formation in order to properly choose these types of hydration promoters in the future.

**Author Contributions:** Conceptualization, B.L. and Y.-Y.L.; validation, B.L. and Y.-Y.L.; formal analysis, Y.-Y.L. and Y.-L.L.; investigation, Y.-Y.L. and Y.-L.L.; resources, B.L.; data curation, Y.-Y.L.; writing—original draft preparation, B.L. and Y.-Y.L.; writing—review and editing, B.L., Y.-Y.L. and Y.-L.L.; visualization, Y.-Y.L.; supervision, B.L.; project administration, B.L.; funding acquisition, B.L. All authors have read and agreed to the published version of the manuscript.

**Funding:** The authors gratefully acknowledge the National Natural Science Foundation of China (51876017) and the Chongqing Innovation Support Funds (cx2020039) for providing financial support for this work.

**Institutional Review Board Statement:** Not applicable.

**Informed Consent Statement:** Not applicable.

**Conflicts of Interest:** The authors declare no conflict of interest.

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
