# Peer review of "A Review of Natural Gas Hydrate Formation with Amino Acids"

_jmse, doi:10.3390/jmse10081134_

Round 1

Reviewer 1 Report

Please, mention what gas is under consideration when you speak about conditions of gas hydrate formation. 

The main problerm in  SNG technology is rapid formation of dense, monolithic hydrate samples, not acceleration of the gas hydrate formation reaction by different additives like amino acids. Reaction velosity is considerably regulated by heat exchange intesity with environment. Different additives can accelerate nucleation and crystal growth, but they are not able to affect reaction velosity as much as heat transfer through reactor walls.

Author Response

Thank you for your particularly useful comments! The reply to the reviewer's comments can be found in the attached file.

Reviewer 2 Report

I have the following suggestions to improve the quality of the article:

- The abstract must be re-written: the first paragraph of it consists of a general state of the art of natural gas hydrates. The abstract should be entirely focused on the research exposed in the following sections of the article.

- The introduction is too long and far from the scope of the article.

Section 2 is not clear. The title is: "Amino acids-promoted hydrate formation systems". However, in this section the authors wrote about promotion, inhibition, contemporary use of amino acids and other additives. The action on the kinetics and on the thermodynamics are not well distinguished here. 

- if writing about the contemporary use of AA and other additives, I think that a more deepened discussion could be performed.

- the diagrams in fig. 4 - 6 seem to habe been plotted by considering only few references; as a consequence of it, the information they contain are not particularly different from what present in the related references. Moreover, the degree of elaboration seems to be low (see for instance Fig. 3: I have found a similar diagram in J. Nat Gas Sci Eng 64 (2019) 52 - 71).

- Section 4.3 and 5 are very close to each other and should be put together;

- The overall structure of the article is, in my opinion, weak and should be revised. More original discussions and personal consideration of the authors should be included in the  manuscript.

Author Response

(The authors gave the same response as above.)

Reviewer 3 Report

Dear authors

I believe that your review paper can be published after major revision. Please find my comments attached.

Author Response

(The authors gave the same response as above.)

Round 2

Reviewer 2 Report

The authors carefully revised the article and improved it significantly, also including all the suggestions I included in the previous revision. 

The article can now be considered for publication.

I suggest to the author to explain in the text which are the properties of amino acids that make them suitable for hydrate promotion/inhibition. In this sense, some useful information can be found in:

Nasir Q, Suleman H, Elsheikh YA. A review on the role and impact of various additives as promoters/inhibitors for gas hydrate formation. J. Nat. Gas Sci. Eng., 76 (2020) 103211

Moreover, Figure 3 is now more appropriate. In my opinion, it could be improved with further data. The authors can find these date in the previous ref and in the following one:

Rossi F, Gambelli AM. Thermodynamic phase equilibrium of single-guest hydrate and formation data of hydrate in presence of chemical additives: a review. Fluid Phase Equilibr., 536 (2021) 112958

Author Response

The response to the reviewer's comments can be found in the attached file.
